# META-IMITATION LEARNING BY WATCHING VIDEO DEMONSTRATIONS

**Jiayi Li**[1,2]**, Tao Lu**[2]**, Xiaoge Cao**[1,2]**, Yinghao Cai**[2] **& Shuo Wang**[1,2,3]
[1]School of Artificial Intelligence, University of Chinese Academy of Sciences
[2]State Key Laboratory of Management and Control for Complex Systems, Institute of Automation, Chinese Academy of Sciences
[3]Center for Excellence in Brain Science and Intelligence Technology, Chinese Academy of Sciences
{lijiayi2019,tao.lu,caoxiaoge2020,yinghao.cai,shuo.wang}@ia.ac.cn

## ABSTRACT

Meta-Imitation Learning is a promising technique for the robot to learn a new task from observing one or a few human demonstrations. However, it usually requires a significant number of demonstrations both from humans and robots during the meta-training phase, which is a laborious and hard work for data collection, especially in recording the actions and specifying the correspondence between human and robot. In this work, we present an approach of meta-imitation learning by watching video demonstrations from humans. In comparison to prior works, our approach is able to translate human videos into practical robot demonstrations and train the meta-policy with adaptive loss based on the quality of the translated data. Our approach relies only on human videos and does not require robot demonstration, which facilitates data collection and is more in line with human imitation behavior. Experiments reveal that our method achieves the comparable performance to the baseline on fast learning a set of vision-based tasks through watching a single video demonstration.

## 1 INTRODUCTION

The demonstration provides skill guidance for specifying robotic tasks. Through it, robots can acquire many complex skills, including table tennis (Mülling et al., 2013), pouring water (Sermanet et al., 2018), and picking objects (Levine et al., 2018). Most prior work assumes that robots can receive demonstrations via kinesthetic teaching (Elliott et al., 2017; Ragaglia et al., 2018), teleoperation (Savarimuthu et al., 2017; Zhang et al., 2018), or crowdsourcing platform (Mandlekar et al., 2018; Wang et al., 2021), which are distinct from the way how humans imitate others. Due to the mirror neurons (Tranel et al., 2003; Molenberghs et al., 2009), humans are often able to watch others act, infer the intention, map it to their own embodiment, expand skill set and enhance the representations of the world. Motivated by this, we aim to endow robots with the ability to learn manipulation skills via video demonstrations from humans without access to the actions of the demonstrator. Then the key challenge we would face is how to bridge the human-robot domain gap caused by the morphological difference and infer the performed skills from the raw video.

One way to overcome the domain gap is to translate the human videos to the robot domain using a generative modeling approach (Sharma et al., 2019; Smith et al., 2019; Yang et al., 2020). While this approach is intuitive, it is prone to capture task-agnostic pixel information so that requires task guidance and it is necessary to evaluate the quality of translated images. The other way is to model visual imitation as a one-shot learning problem (Duan et al., 2017), which can be solved with meta-learning algorithms (Finn et al., 2017b; Yu et al., 2018b). Domain gaps can be addressed with a supervised training process through the cross-domain dataset. However, meta-learning methods often require large amounts of demonstrations both from humans and robots during the meta-training phase, which is a demanding and time-consuming process for data collection, especially in recording the actions and specifying the correspondence between human and robot.

In this paper, we introduce a new approach of meta-imitation learning by watching video demonstrations from humans. This approach automatically translates human videos to robot demonstrations using a new-designed generative model and trains the meta-policy in the imagined compact latent space learned during the translation process from human to robot. Empirical results suggest that the learned meta-policy achieved the comparable results to the baselines on learning a new set of chal-

lenging vision-based tasks under the condition of watching one video demonstration. This is the first time, to our knowledge, that meta-imitation learning has been addressed only by using image observation with no need for demonstration information of robot states and actions. Our approach learns the correspondence between human and robot and trains end-to-end, which places the least demand on the imitation system and is a natural learning way just like humans. The key contributions of this paper are summarized as follows:

- Learning action-awareness compact latent representation from human to robot. We proposed a novel CycleGAN structure, named A-CycleGAN, that automatically translates the human domain data into the robot domain and learns an action-awareness compact latent representation with parallel training of inverse dynamic model under the robot domain. This new structure helps us to achieve effective robot demonstration data from human videos.

- Self-adaptive meta-imitation learning under the imagined latent space. The meta-policy is trained with the learned latent states achieved from A-CycleGAN and the actions predicted from the learned inverse dynamic model. All data utilized during training could be regarded as being generated by imagination based on human demonstrations. A self-adaptive mechanism is presented to evaluate the quality of the translated data and to prioritize high-quality data for better gradients update, ensuring that the meta-policy is properly trained even when the data does not come from actual robot actions.

- Empirical performance on challenging vision-based tasks. We pair our approach with existing meta-imitation learning methods which learns the meta-policy with both human and robot demonstrations on a set of challenging tasks. With the same human video demonstrations, our method achieved comparable results to the baselines in one-shot new task learning.

## 2   RELATED WORK

**Learning from Demonstration (LfD).** LfD is a rich and diverse research field that focuses on allowing robots to learn skills from humans or other expert demonstrations(Ravichandar et al., 2020). We follow the problem of learning new task from one video demonstration of human, conforming to the reality that the robot can not receive the action information from the demonstration directly. The key point is how to specify the correspondence between human and robot. It is usually resolved by two types of methods, either explicit or implicit. Many explicit methods manually define how human activities are translated into robot actions, with the help of specific visual recognition pipelines (Nguyen et al., 2018; Yang et al., 2015). These techniques rely on the precise detection of hands and objects. Recent works are mainly concentrated on representation learning techniques implicitly. Many of these produce metric embeddings that focus on the interactions in the environment and are insensitive to nuisance variables such as viewpoint and appearance in an unsupervised fashion (Sermanet et al., 2018; Schmeckpeper et al., 2020; Zhou et al., 2021). Other approaches train action or task classifiers from human and robot videos. The distance between the observed demonstration and the execution of the robot is used as a reward of reinforcement learning in learning the demonstrated task (Sermanet et al., 2016; Sieb & Fragkiadaki, 2018; Shao et al., 2020a; Pauly et al., 2021). There are also ways to align the two domains using the generative model. Some work generates robot domain images by integrating human-robot paired data with contexts (Liu et al., 2018; Sharma et al., 2019), while others employ pixel-level image translation with unpaired training data (Smith et al., 2019; Xiong et al., 2021).

**One-shot Imitation Learning.** Humans naturally acquire the ability to execute a new task by witnessing it once performed by another individual. Prior research has enabled robots to acquire similar ability. Some methods take advantage of the attention mechanism of the model or symbol planner to extract the operated items from a demonstration and transfer the skill directly (Huang et al., 2019; Shao et al., 2020b; Dasari & Gupta, 2020). Other approaches learn a compact representation of a task joint with the control policy and deduce information from unfamiliar tasks in task-embedded space (James et al., 2018; Bonardi et al., 2020). In other work, the one-shot learning problem is solved with model-agnostic meta-learning (MAML) algorithm (Finn et al., 2017a). It learns the task initial parameters by leveraging large amounts of prior meta-training data and optimize it via one or a few steps of gradient descent with new tasks (Finn et al., 2017b). Domain Adaptation Meta-Learning (DAML) (Yu et al., 2018b) is an extension of MAML and could learn robotic manipulation skills from a single video of human. And it is also extended on multi-stage manipulation tasks (Yu et al., 2018a). Our approach is based on this technique, but we recover the necessary information

to complete the meta-policy learning from human videos without specialized access to the robot demonstration (such as action and state). This greatly lowers the difficulties in data collections for policy-training and places least demand on the imitation system, which is a more natural way for robot imitation learning.

## 3 PRELIMINARIES

To learn from the human demonstration, our approach builds upon some prior work and presents an extension of algorithm that relaxes its implementation requirements. In this section, we will overview unsupervised image-to-image translation (Zhu et al., 2017) and domain-adaptive meta-learning (DAML) algorithm (Yu et al., 2018b).

### 3.1 UNSUPERVISED IMAGE-TO-IMAGE TRANSLATION

We treat the human-to-robot observation translation as an unsupervised image-to-image translation problem. That is, we aim to map images from a source domain $X$ (e.g., human domain) to a target domain $Y$ (e.g., robot domain) in the absence of paired examples. CycleGAN (Zhu et al., 2017) is a technique for learning a mapping between two image domains from unpaired data. Mathematically, CycleGAN learns the translator $G : X \rightarrow Y$ and inverse translator $F : Y \rightarrow X$ given training samples $\{x\} \in X$ and $\{y\} \in Y$. The image is embedded by the encoder of the translator into the latent space and then reconstructed by the decoder. The embedding in the latent space is domain-agnostic. The generator needs to be jointly trained with two adversarial discriminators $D_X$ and $D_Y$, where $D_X$ attempts to distinguish between images $\{x\}$ and translated images $\{F(y)\}$ to encourages $F$ to generate the outputs indistinguishable from domain $X$, and vice versa for $D_Y$ and $G$. The model is trained by minimizing adversarial losses $L_{GAN}$:

$$L_{GAN}(G, D_Y, X, Y) = \mathbb{E}_{y \sim p_{data}(y)}[\log D_Y(y)] + \mathbb{E}_{x \sim p_{data}(x)}[\log(1 - D_Y(G(x)))], \quad (1)$$

where $G$ aims to minimize this objective while an adversary $D$ strives to maximize it. It is similar for $F$ and $D_X$. Further to avoid mode collapse that all input images map to the same output image, the cycle consistency loss $L_{cyc}$ is introduced to incentivize the original image recovered after two translators:

$$L_{cyc}(G, F) = \mathbb{E}_{x \sim p_{data}(x)}[||F(G(x)) - x||_1] + \mathbb{E}_{y \sim p_{data}(y)}[||G(F(y)) - y||_1]. \quad (2)$$

Combining the two losses yields the full objective for translation:

$$L_{CG}(G, F, D_X, D_Y) = L_{GAN}(G, D_Y, X, Y) + L_{GAN}(F, D_X, X, Y) + \lambda L_{cyc}(G, F), \quad (3)$$

where $\lambda$ is the hyperparameter that controls the relative importance of the two objectives.

### 3.2 DOMAIN-ADAPTIVE META-LEARNING

Domain-adaptive meta-learning (DAML) is an extension of the model-agnostic meta-learning algorithm (MAML), which could learn new task from one human video demonstration. Assuming that meta-training and meta-test tasks are drawn from the same distribution $p(\mathcal{T})$, MAML is a meta-learning algorithm that intends to learn new tasks using a small amount of data. It extracts prior task knowledge using meta-training tasks, then inferences under learned priors from the meta-test demonstration (Grant et al., 2018). Specifically, consider a supervised learning problem with a loss function denoted as $L(\theta, D_{\mathcal{T}})$, where $\theta$ denotes the model parameters and $D_{\mathcal{T}}$ denotes the labeled data for task $\mathcal{T}$. During meta-training, MAML samples a task $\mathcal{T}$ and data from $D_{\mathcal{T}}$, which are randomly partitioned into two sets, $D_{tr}$ and $D_{val}$. MAML optimizes the model parameters $\theta$ such that one or a small number of gradient steps on $D_{tr}$ will produce maximally effective behavior on $D_{val}$, corresponding to the following objective:

$$\min_{\theta} \sum_{\mathcal{T} \sim p(\mathcal{T})} L(\theta - \alpha \nabla_{\theta} L(\theta, D_{\mathcal{T}}^{tr}), D_{\mathcal{T}}^{val}) = \min_{\theta} \sum_{\mathcal{T} \sim p(\mathcal{T})} L(\phi_{\mathcal{T}}, D_{\mathcal{T}}^{val}), \quad (4)$$

where $\alpha$ is the step size of the gradient update and the $\phi_{\mathcal{T}}$ is the updated parameters. The meta-optimization is performed over the model parameters $\theta$, whereas the objective is derived using the updated parameters $\phi_{\mathcal{T}}$. At meta-test period, MAML executes gradient descent with $\theta$ to adapt the new task $\mathcal{T}_{test}$ using examples drawn from it:

$$\phi_{\mathcal{T}_{test}} = \theta - \alpha \nabla_{\theta} L(\theta, D_{\mathcal{T}_{test}}^{tr}). \quad (5)$$

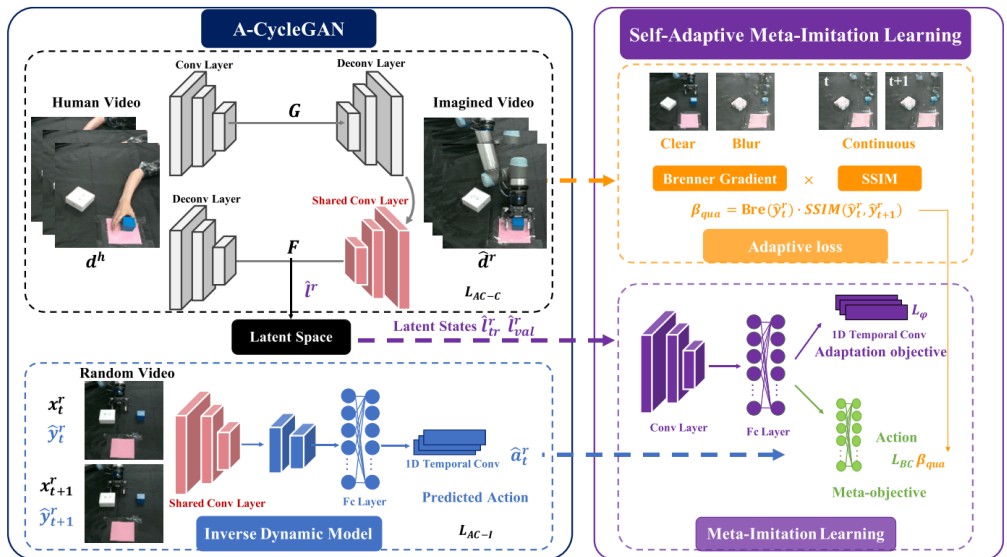

Figure 1: **Illustration of the Overall Architecture.** Left: Action-awareness CycleGAN (A-CycleGAN): The image translation model establishes the bidirectional mapping between the human demonstrations and the imagined videos of the robot. Then, the inverse dynamic model takes as input the latent state to predict the corresponding robot action $\hat{a}^r$. The CycleGAN part and Inverse Dynamic Model part share the encoder layers of generator $F$ (shown in pink). Right: Self-Adaptive Meta-Imitation Learning. The Meta-Imitation Learning structure is similar to DAML and the meta-policy is trained with the data from the latent states $\hat{l}^r$ and predicted actions $\hat{a}^r$. The translated imagined robot videos $\hat{d}^r$ are evaluated by $\beta_{qua}$ to adaptively adjust the meta-objective.

DAML applies the MAML algorithm to the domain-adaptive one-shot imitation learning setting to solve the problem of learning from human video. Unlike the supervised meta-learning issue, it can not use the standard imitation learning loss for the inner adaptation objective computed using $D_{tr}$ without the human's actions. To handle the domain shift between the human and robot, DAML adopts the multiple layers of 1D convolutions over time to donate the adaptation objective $L_\psi$, which can be interpreted as simply directing the policy parameter update to modify the policy to pick up on the right visual cues in the scene. During meta-training, the meta-objective for $\theta$ is the behavioral cloning loss $L_{BC}$:

$$\min_{\theta, \psi} \sum_{\mathcal{T} \sim p(\mathcal{T})} \sum_{d^h \in D_\mathcal{T}^h} \sum_{d^r \in D_\mathcal{T}^r} L_{BC}(\theta - \alpha \nabla_\theta L_\psi(\theta, d^h), d^r), \tag{6}$$

where $d^h$ is the human demonstration and $d^r$ is the robot demonstration. After training, DAML runs the gradient descent using the adaptation loss $L_\psi$ based on the human demonstration for a new task.

## 4 METHOD

Our goal is to learn the task prior knowledge and rapidly adapt to new tasks using only human video without robot demonstration. Our method comprises two main modules: domain adaptation A-CycleGAN and self-adaptive meta-imitation learning module with adaptive loss. In this section, we will systematically describe them.

### 4.1 A-CYCLEGAN: ACTION-AWARENESS CYCLEGAN FOR DOMAIN ADAPTATION

To extract the available states of the robot domain from human videos $d^h \in D_\mathcal{T}^h$, including the domain-agnostic but action-awareness representations of scenes and corresponding actions for robot, we propose a new domain-adaptation method, named Action-awareness CycleGAN (A-CycleGAN) to establish the bidirectional mapping between the human demonstrations (human/source domain) and the corresponding generated robot videos (robot/target domain). In meta-imitation phrase, we refer to the robot videos translated from the human demonstrations by the A-CycleGAN as the imagined robot videos $\hat{d}^r \in \hat{D}_\mathcal{T}^r$. Given that the original CycleGAN only supervise the overall

content and style in the scene which may be inclined to reconstruct fixed task-agnostic background, we combine it with a coupled inverse dynamic controller $\pi_I(\cdot)$ to predict the action-awareness images of human behavior in the robot domain to propel the task-related translation. So in general, A-CycleGAN includes two parts: the CycleGAN part and the Inverse Dynamic Model part. For CycleGAN part, due to the multi-task experiment setting, the original objective of CycleGAN stated in Section 3.1 can not guarantee the mapping consistency within the scene, for which we collect a limited number of pairwise human and robot random video $p^h$, $p^r$ (these pairwise videos do not have to be perfectly aligned and easy to collect) and add an auxiliary objective as follows:

$$L_{aux}(G, F) = E_{p^r, p^h}[||G(p^h) - p^r||_1 + ||F(p^r) - p^h||_1]. \tag{7}$$

That is, the output of the generator is constrained by the other domain image, improving the certainty of the content in the scene.

As for inverse dynamic model, it takes as input current visual observation of the robot $x_t^r$ along with the observation for the next time step $x_{t+1}^r$ to predict the actions $a_t^r = \pi_I(x_t^r, x_{t+1}^r)$ the robot should take to make the transition to its next state. The training data is sampled from robot random move video $(x_1^r, a_1^r, \cdots, x_T^r, a_T^r)$, in which the robot moves around in the scene but does not specifically attempt the task. So the full objective of A-CycleGAN is:

$$
\begin{aligned}
L_{AC-C} &= L_{GAN} + L_{aux} + \lambda L_{cyc}, \\
L_{AC-I} &= ||\pi_I(x_t^r, x_{t+1}^r) - a_t^r||^2
\end{aligned}
\tag{8}
$$

The two parts of A-CycleGAN share the network structure of encoding layers of generator $F$, as depicted in Figure 1. During training, they are trained in parallel. Through the shared network, the gradient update of the inverse dynamic model would guide the CycleGAN to retain the action-awareness information in the translated images, which is critical for ensuring the translated data is useful for the subsequent meta-imitation learning.

After A-CycleGAN training, the human images could be translated into robot domain, and the generated robot images are used to predict the corresponding actions through inverse dynamic model. These pairs of robot images and actions could be the training data for meta-policy learning. Here we refer to the generated robot images as Imagined Robot Images $\hat{y}^r$, the sequences of $\hat{y}^r$ as Imagined Robot Video $\hat{d}^r$, and the corresponding predicted actions as Imagined Robot Action $\hat{a}^r$. Considering the quality of the translated images and the high-dimension property, in our work we do not directly employ $\hat{y}^r$ for the following meta-learning but rather the latent states $\hat{z}^r$ outputted from the encoding layers of generator $F$. Here we refer $\hat{z}^r$ as Imagined Latent States and the sequences of $\hat{z}^r$ as Imagined Latent Video $\hat{l}^r$ (as stated in Figure 1). Latent states are domain-agnostic and provide a more stable representation of image demonstrations. Our empirical results verified that the meta-policy achieved better performance based on $\hat{z}^r$ than $\hat{y}^r$.

## 4.2 Self-Adaptive Meta-Imitation Learning

Through Imagined Robot Video $\hat{d}^r$ and Imagined Robot Action $\hat{a}^r$, our algorithm can work on the imagined data and does not require a dataset of real robot demonstration during training, which would greatly reduce the burden on kinesthetic teaching or teleoperation.

However, we can not blindly take these data as a precise input to the model, as the quality of the translated images produced by the generative model in different scenes varies, and some may produce artifacts. Thus we suggest evaluating the data quality to receive the learning worth adaptively. In practice, $\hat{z}^r$ are encoded by $\hat{y}^r$ translated through the human demonstration, so we evaluate it based on the quality of $\hat{y}^r$. With the principle that the imagined video should be as clear and continuous as the real video, we adopt the Brenner gradient to detect edges in the image (Ali & Mahmood, 2018) and the structural similarity index measure (SSIM) to assess the continuity of adjacent images in the video (Wang et al., 2004). Mathematically, the Brenner gradient $Bre(\hat{y}_t^r)$ computes the first difference between a pixel and its neighbor with the horizontal and vertical distance of 2. The SSIM considers comparison measurements of luminance, contrast and structure between the two images of $\hat{y}_t^r$, $\hat{y}_{t+1}^r$:

$$SSIM(\hat{y}_t^r, \hat{y}_{t+1}^r) = \frac{\left(2\mu_{\hat{y}_t^r}\mu_{\hat{y}_{t+1}^r} + c_1\right)\left(2\sigma_{\hat{y}_t^r\hat{y}_{t+1}^r} + c_2\right)}{\left(\mu_{\hat{y}_t^r}^2 + \mu_{\hat{y}_{t+1}^r}^2 + c_1\right)\left(\sigma_{\hat{y}_t^r}^2 + \sigma_{\hat{y}_{t+1}^r}^2 + c_2\right)},$$ 

(9)

where $\hat{y}_t^r$ is the image at step $t$ in $\hat{d}^r$. $\mu_{\hat{y}_t^r}$ is the average of the image. $\sigma_{\hat{y}_t^r}^2$ is the variance. $\sigma_{\hat{y}_t^r\hat{y}_{t+1}^r}$ is the covariance of two images. $c_1$ and $c_2$ are two variables to stabilize the division with a weak denominator. SSIM index is a decimal value between -1 and 1, with 1 being reached only when two identical sets of data are present. We normalize both the Brenner gradient and the SSIM to the range of 0 to 1. In this way, the quality of the image is denoted as $\beta_{qua}$:

$$\beta_{qua}(\hat{y}_t^r) = Bre(\hat{y}_t^r) \cdot SSIM(\hat{y}_t^r, \hat{y}_{t+1}^r).$$ 

(10)

This metric is applied to adaptively adjust the learning loss of the imitation module.

As discussed in Section 3.2, we separate the $\hat{l}^r$ into two sets, $\hat{l}_{tr}^r$ and $\hat{l}_{val}^r$. Considering the quality loss of $\hat{l}^r$ and predicted actions $\hat{a}^r$ to the real demonstrations, we do not directly use MAML but DAML structure for meta-imitation learning and keep the inner objective $L_\psi$ in the form of 1D convolutions over time during meta-training phase to lower the error induced by $\hat{a}^r$. Thus, $\hat{l}_{tr}^r$ are the sequences of states $(\hat{z}_1^r, \cdots, \hat{z}_T^r)$ and $\hat{l}_{val}^r$ includes sequences of states and actions $(\hat{z}_1^r, \hat{a}_1^r \cdots, \hat{z}_T^r, \hat{a}_T^r)$. The meta-objective is the normal behavior cloning loss $L_{BC}$ with the adaptive adjustment:

$$\min_{\theta,\psi} \sum_{\hat{l}_{tr}^r} \sum_{\hat{l}_{val}^r} \beta_{qua} L_{BC}(\theta - \alpha\nabla_\theta L_\psi(\theta, \hat{l}_{tr}^r), \hat{l}_{val}^r).$$ 

(11)

Compared to the common meta-imitation learning module discovering the visual cues to accomplish the task only through action guidance, which may result in insufficient adaptation to the new scene, our method based on the latent states and adaptive loss maintains more task-related and high-quality information to facilitate the meta-policy learning. The whole procedure is illustrated in Figure 1.

## 5 EXPERIMENTS

The aims of our experimental evaluation are to answer the following questions:

1. Can our approach allow the robot to learn new skills by observing only one human demonstration without any robot instruction during training?
2. How does our approach compare to the mainstream meta-imitation baseline that requires robot demonstrations?
3. In A-CycleGAN, what is the impact of coupled training the generative model and the inverse dynamic model?
4. How important is it that the meta-imitation module takes the latent states as input?
5. What benefits do we gain from the adaptive loss?

Our task is challenging: it lacks a significant amount of accurate information about the skills throughout the training. Especially the latent states and the corresponding actions used for meta-policy training are all estimated. In our evaluation, we design two robot manipulation skills: shape-drawing and pushing. These two skills are also challenging: The shape-drawing skill needs the robot to follow the designated shape lines accurately, which has not appeared in prior meta-learning experiments. The pushing skill needs robot to push the target object to the goal under the new distractor (different from demonstrations) in the meta-test phase, which would be more effective in verifying the generalization of the meta-policy. It is also harder than the assignment in DAML (Yu et al., 2018b) in which the robot only needs to push the target away under the same distractor as in demonstrations. To our knowledge, we are the first to propose the meta-imitation learning method using only human video demonstrations during meta-training (referred to as **MILV**). Consequently, we compare our method with the meta-imitation learning baseline **DAML** which could learn the new task from a single video but require robot demonstration during meta-training. We also compare with our ablations to understand the complementary performance of each module, which includes decoupled training of the generative model and the inverse dynamic model (referred to as **DeGI**); directly feeding the meta-imitation module with the imagined robot images (**FeedImg**), and updating the meta-imitation module with the stable loss (**WStable**).

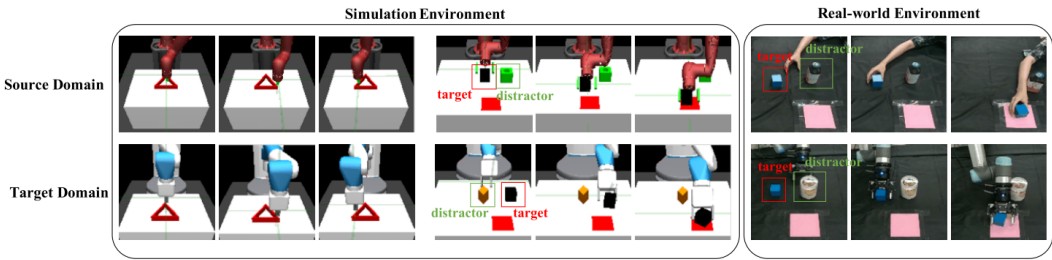

Figure 2: Examples of simulated drawing (left), pushing (middle), and real-world pushing (right) tasks. The top row shows the source domain demonstration, while the bottom shows policy execution in target domain.

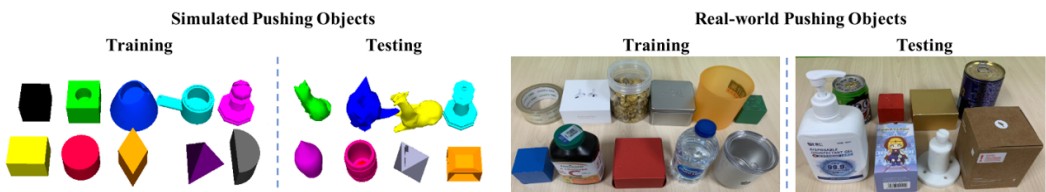

Figure 3: All of the objects used for training and evaluation for pushing task.

## 5.1 SIMULATED TASKS

To facilitate the evaluation, we first consider performing the method entirely in the simulated environment where one robot learns skills from another. As illustrated in Figure 2, The experiment involves the shape-drawing task, depicting the shape following the trajectory instructions; object pushing task, pushing the specified object into the target amid one distractor. We design the environment starting from the OpenAI Gym environment (Brockman et al., 2016), based on the simulated MuJoCo physics engine (Plappert et al., 2018; Todorov et al., 2012).

The demonstrator is a 7-DoF Sawyer robot (source domain) with a two-finger gripper, and the learner is a 7-DoF Fetch robot (target domain) with the different appearance. The camera records sequences of 128 × 128 RGB pixels images from a different perspective. The policy will control the gripper incremental movement. For the drawing task, the goal is to depict the shape according to the red trajectory indication with a random start position. We selected 11 different shapes, including triangle, circle, and square, et al. with various sizes for meta-training. The training set consists of 1,320 Sawyer robot demonstrations (source domain), an equal number of Fetch robot random videos (target domain), and 66 Sawyer-Fetch paired videos. For DAML, Fetch robot random videos are replaced by the equal number of demonstrations corresponding to the Sawyer (including images and actions), and it does not require paired data. For evaluation, we used 12 new shapes including scale and rotation variants of shapes in training set, and unseen new shapes. See Appendix A for more details about the experimental setup and choices of hyperparameters. At test time, the Fetch robot receives one video demonstration per task and evaluates 50 trials. Each trial is 50 steps and we define success when the longest common subsequence length (Bergroth et al., 2000) between the shape drawn by the robot and the real path are longer than the threshold.

As for the pushing task, the goal is to push a particular object with a random start position into the random red target amid the other distractor. The experimental objects have a wide range of shapes, sizes, frictions and masses. Note that the target object in demonstrations is the same in scenes for the adaptation policy, but the distractor is not always consistent. There are 10 objects for the meta-training and 8 for evaluation, including tetrahedron, kettle, pawn, and so on. The datasets contain 1,000 Sawyer demonstrations, an equal number of Fetch random videos, and 60 Sawyer-Fetch paired videos. For DAML, the Fetch random videos are replaced by the equal number of demonstrations (including images and actions) and it does not require paired data. All of the objects are shown in Figure 3.

The testing result is rendered in Figure 4 and the success rates of all experiments are summarized in Table 1, which confirms that our proposed method achieves compelling performance in both tasks.

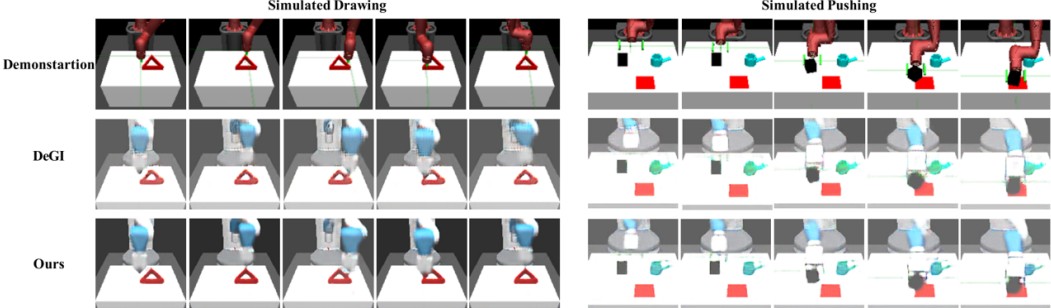

Figure 4: All simulated testing visualization results. From left to right: a demonstration from the source domain, the policy execution in target domain. Note that the distractor is inconsistent for demo and executed scenes in pushing tasks.

Table 1: One-shot success rate in all experiments

| Method | Simulated Drawing | Simulated Pushing | Real-World Pushing |
|---|---|---|---|
| DAML | 43.3% | 67.3% | **58.8%** |
| DeGI | 24.2% | 58.5% | 32.1% |
| FeedImg | 27.1% | 57.6% | 38.1% |
| WStable | 40.0% | 64.6% | 51.3% |
| **MILV(Ours)** | **46.3%** | **69.2%** | 56.3% |

Figure 5: Translated images comparison on the simulated drawing and pushing task.

In simulated drawing task, all methods have the success rate under 50% and are not very high. The main reason is that the image observation is not efficient enough as the robot state, such as joint angles, and the task is also a tough one to complete. In this condition, Ours (46.3%) has the similar performance with DAML (43.3%), even better. The success rate of WStable (40.0%) is slightly lower than ours, indicating that the quality evaluation of the data contributes to the accuracy. DeGI (24.2%) has only half of ours, which means the A-CycleGAN is critical in our methods in translating human videos into the effective action-awareness robot demonstration data. As shown in Figure 5, the translated images obtained by the coupled model (Ours) concentrate more on reconstructing the robotic arm motion and objects in the scene forcing to preserve information required for the meta-imitation module. On the contrary, the images outputs by the decoupled model (DeGI) have no obvious preference and it performs inferior to ours. FeedImg (27.1%) also performs poorly, which means the imagined latent states are more useful features than the imagined robot images for the meta-policy training.

In the simulated pushing task, the whole performance is higher compared to the shape-drawing task. Mainly because the pushing task only cares about whether the goal is in the target area, no matter what kind of path the robot experienced. Our method (69.2%) performs slightly superior than the

Table 2: Breakdown of the failure modes

| Method | Identify Objects | Control |
|---|---|---|
| DAML | 76.9% | 23.1% |
| FeedImg | 48.9% | 51.1% |
| **MILV(Ours)** | 43.4% | 56.6% |

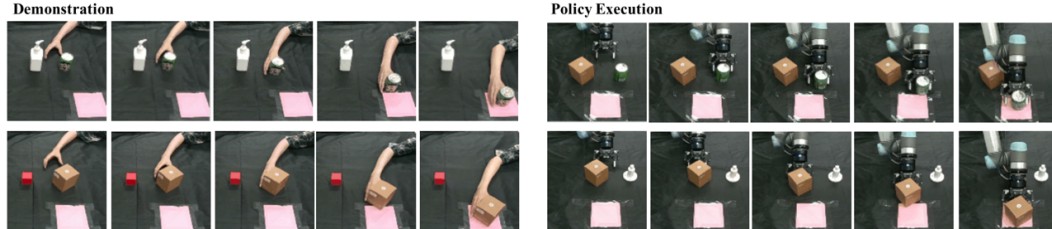

Figure 6: The real-world testing visualization results.

DAML (67.3%). DeGI (58.5%) and FeedImg (57.6%) are worse than WStable (64.6%), which also indicates that A-CycleGAN and the latent states for meta-training are the key modules in our method, and the adaptive loss further improve the performance through high-quality data selection.

In order to further understand the effect of our method, we count the failure mode and its proportion in Table 2, which includes incorrect object identification and control failures. It is apparent that algorithms with image inputs, such as DAML, FeedImg, have a greater likelihood of misidentifying objects than ours, implying that they share the critical flaw: meta-policy supervised only by executed action may be unable to comprehend the correct visual information in the scene, while the latent states learned by A-CycleGAN incorporate more visual information except corresponding to the actions. That means our method could be scaled to broader distributions of tasks. We can also see that compared to DAML, our method has higher control failures (56.6% vs. 23.1%). For control accuracy, we could improve the performance by discovering more features to assist domain alignment in the next-step work.

## 5.2 REAL-WORLD PUSHING

We designed the real-world pushing environment using a UR5 robotic arm and RGB camera. The goal is to push the correct object to the pink target area. The sample videos are shown in Figure 2. Images taken from the human and robot domains have unique perspectives. For meta-training, we collected a dataset with 11 objects, consisting of 550 human demonstrations, 550 robot random videos and 110 human-robot paired videos. For DAML, random videos are replaced by 550 UR5 demonstrations corresponding to humans (include images and actions). During the evaluation, we provided 8 novel objects. The task is considered successful if the robot pushes the target object within the center of pink region. We tested each of the objects 20 times.

The testing results in the third column of Table 1 report the success rate of our real-world experiments. The whole success rate (lower than 60%) is lower than in simulation environment, which is mainly influenced by the observation and action noise in real-world. Our method (56.3%) still achieves the comparable results to DAML (58.8%). The performance of DeGI(32.1%) and FeedImg (38.1%) is down greatly and again verifies the two modules of our method are indispensable. Figure 6 shows the real-world testing visualization results. Additional experiments testing results refer to Appendix B.

## 6 CONCLUSIONS

We provide a meta-imitation approach that enables robots to learn the meta-policy by only using video demonstrations from humans and have the ability of one-shot learning new tasks from watching a single video. Our approach automatically translates human videos into robot demonstrations using a new-designed generative model A-CycleGAN and train the meta-policy in the imagined compact latent space with the proposed adaptive loss. Empirical results suggest that the learned meta-policy achieves compelling results to the baselines in one-shot vision task learning both in simulation and real-world. Our approach relaxes the demands for training data, requiring only human video demonstrations and gives a try to the natural imitation learning of robots.

In the future, we hope to improve our task performance by incorporating reinforcement learning techniques for online trial and error. Since our method requires some domain-aligned data, we expect that the content translation model, such as UNIT (Liu et al., 2017), would likely extend our method to adopt human or robot demonstration data from different sources.

ACKNOWLEDGMENTS

We'd like to begin by acknowledging the students and collaborators at CASIA who gave valuable feedback which made the final paper much stronger. In particular, we thank Weiyan Guo, Congjia Su, Junhang Wei, Shaowei Cui and Yaozhong Cao. This work was supported by grants from the National Key R&D Program of China(grant 2019YFB1311901).

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

## A    ADDITIONAL EXPERIMENTAL DETAILS

We provide additional experimental details for all experiments, including data collection, evaluation, and training hyperparameters.

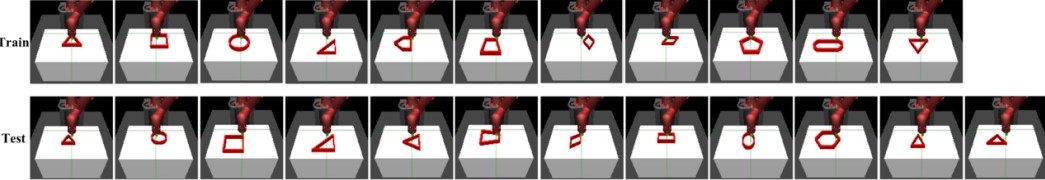

Figure 7: The display of all training and testing shapes for the drawing task. For testing shapes, the first four are scale variants of the training set, the middle three are rotation variants of the training set, and the last five are novel unseen shapes.

## A.1 EXPERIMENTAL SETUP

In the experiment, the input to the policy is only sequences of $128 \times 128$ RGB pixels images, without any information on the robot joint or end-effector. The policy output is the incremental movement of the robot end-effector in 3D space, as $(\Delta x, \Delta y, \Delta z)$, where $\Delta x \in [0, 5\text{cm}]$, $\Delta y \in [0, 5\text{cm}]$, and $\Delta z = 0\text{cm}$ ($z$ is set as a constant 0.3cm above the table). For shape-drawing tasks, all shapes to be depicted are shown in Figure 7. The policy roll-out is considered successful if the longest common subsequence lengths of the passed trajectory and the real shape are greater than 30 (60% of the total length). For the pushing task, as illustrated in Figure 4, only the target object is identical across the robot demonstrations and the actual execution scenes; the distractor is not necessarily the same. The push is considered successful if the target object lands on the target area at the last time step. In simulation, the target area is the red rectangle area(as indicated in Figure 2) with 12cm width and 12cm height. In real-world, the target area is the pink rectangle area(as indicated in Figure 2) with 12cm width and 12cm height . The expert policy inputs include the position and size of the shape for the drawing task, as well as the position of the target object, the robot end-effector and the target for the pushing task. We develop the corresponding solutions for each task using artificially formulated rules.

The policy is fed just the sequence of $128 \times 128$ RGB pixels images from a RealSense D455 camera, and the length of each trajectory is limited to 50 steps. A trial is considered a success if the center of the target object lands on the target within 50 step. To collect demonstration data for one task, we randomly assign a target location for each trial and select a target object and a random distractor at a random location in front of the robot. The robot demonstrations provided to DAML for training are generated by rules.

## A.2 HYPERPARAMETERS FOR NETWORKS

For all experiments, our image translation model uses a convolutional neural network where the encoder uses 3 convolutional layers with 16 7×7, 32 3×3, 64 3×3 filters respectively, followed by four residual modules (each includes two convolutional layers), and the decoder contains deconvolutional layers restoring the image to the original size. The discriminator $D_X$ and $D_Y$ uses 3 strided convolutions and 2 non-strided convolutions, which aim to classify whether overlapping image patches are real or fake. Its architecture can work on images of any size in a fully convolutional fashion (Zhu et al., 2017). We set $\lambda = 10$ and the batch size is 1 in all experiments. These networks are trained from scratch with a learning rate of 0.0002.

The inverse controller takes latent states from the generator $F$ of CycleGAN as the input, using a strided convolution and a non-strided convolution with 16 5×5 filters, followed by a spatial softmax and 2 fully-connected layers with 200 hidden units. The training loss is the mean squared error between the predicted actions and the ground truth robot command. Throughout the training, the learning rate remains constant at 0.001.

The meta-imitation module also takes latent states of the translation model as the input, connecting with 2 strided convolutions and a non-strided convolution with 16 5×5 filters, followed by a spatial softmax and 3 fully-connected layers with 200 hidden units. The last layer applies the two-head architecture, with one head used for the pre-update demonstration using a fully-connected layer and one head used for the post-update policy consisting of three layers of temporal convolutions, the first two with 10×1 filters and the third with 1×1 filters. Our ablation experiments with image input

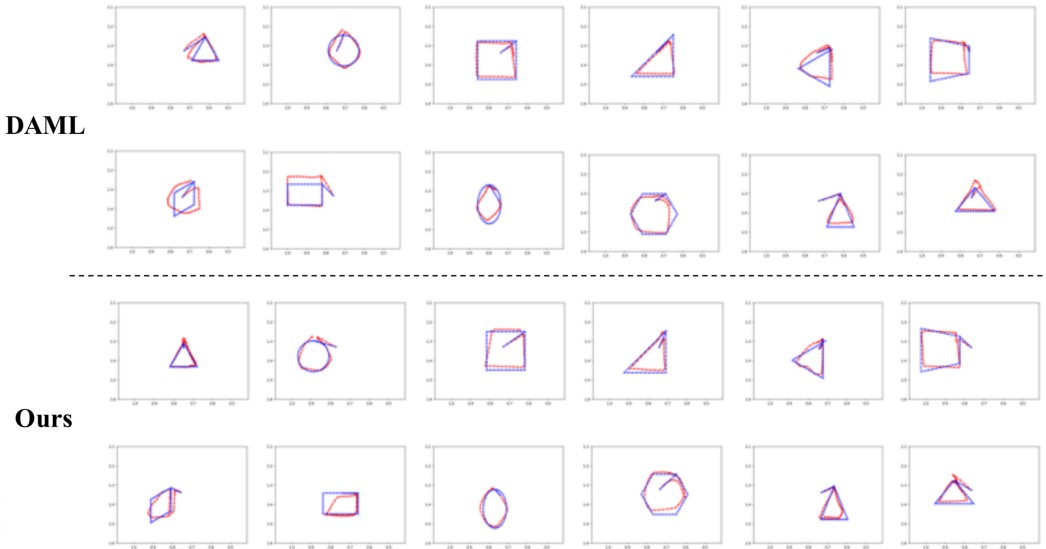

Figure 8: The visualization testing results of our method and DAML in the shape-drawing task.

take human and translated images as inputs. The network structure of the meta-imitation module comprises 6 convolutional layers, as well as the same fully-connected layers and others as described above. The policy used 1 meta-gradient update with the step size $\alpha = 0.01$.

# B ADDITIONAL EXPERIMENTAL RESULTS

## B.1 SIMULATED DRAWING

The visualization testing results of our method and DAML in the shape-drawing task are shown in Figure 8, with blue trajectory representing the groundtruth and red representing the policy execution. It can be seen that both methods effective for the majority of the novel shapes. However, for some shape, such as the first row fifth and the second row first, our method do not perfectly fit the groundtruth but close to it, while the DAML is considerably different from it, preferring to draw these two as arrow shapes in the training dataset. The same condition appears in the second row second which is likely to be drawn as a square. As a result, we suggest that DAML falls short of generalization for new shapes, whereas errors introduced by the translation and inverse dynamic model in our technique prevents model over-fitting to a certain extent.

## B.2 REAL-WORLD PUSHING

Failures in real-world pushing tasks are presented in Figure 9. Due to the more complex environment and misaligned paired data, the task is more likely to fail in real-world conditions than in the simulation settings.

## B.3 ADDITIONAL REAL-WORLD PUSHING EXPERIMENT

Table 3: Results on real-world pushing forward task

| Method | Success Rate | Failure Modes | |
| --- | --- | --- | --- |
| | | Identify Object | Control |
| DAML | 81.3% | 86.7% | 13.3% |
| **MILV(Ours)** | 77.5% | 63.9% | 36.1% |

To further validate the performance of our method compared with DAML, we run an additional real-world pushing experiment designed as in the DAML paper (Yu et al., 2018b). As illustrated in

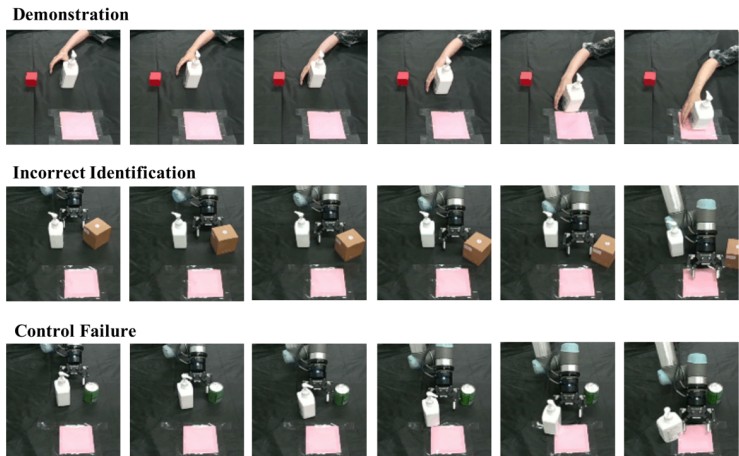

Figure 9: Instances of failure in real-world pushing tasks.

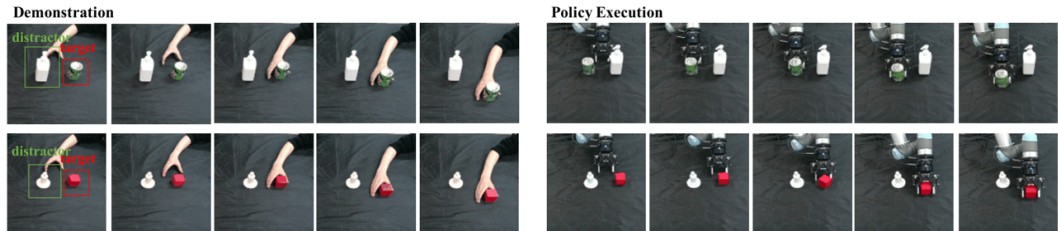

Figure 10: The demonstration and testing results on pushing forward task.

Figure 10, the goal of task is just to push the target object forward for more than 5cm amid the same distractor as that in the demonstration. Apparently, this task is much easier than the pushing task that we designed in our work. Other experimental settings are the same as ours, including the type of objects, the perspectives of human and robot, the number of training data and so on.

Figure 10 shows the visualization results. As demonstrated in Table 3, our testing success rate is 77.5% and DAML is 81.3%, which reveals the promising performance of our method. We count the failure mode and again find our method provides a more accurate identification of the target object.

### B.4    ADDITIONAL TRANSLATED RESULTS

Here we provide some qualitative examples of the translated videos using our translation model A-CycleGAN in Figure 11. The odd lines are demonstrations from the source/human domain, even are translated videos.

### B.5    THE CORRELATION BETWEEN THE ADAPTIVE METRIC AND THE TRANSLATION ERROR

Here we measure the actual translation error $e$ by computing the SSIM of the paired corresponding video and analyze the correlation between the adaptive metric $\beta_{qua}$ and $e$ on simulated pushing and real-world pushing tasks.

As shown in Figure 12, the $\beta_{qua}$ is positive correlated with $e$, which is consistent with our intuition.

### B.6    THE EFFECT OF THE NUMBER OF THE PAIRED VIDEO

We evaluate our method on a range of paired video numbers and show the test success rate on simulated pushing task in Figure 13. Apparently, the increase in the number of paired data means that the increase of supervised data for translation model can improve the test success rate. When

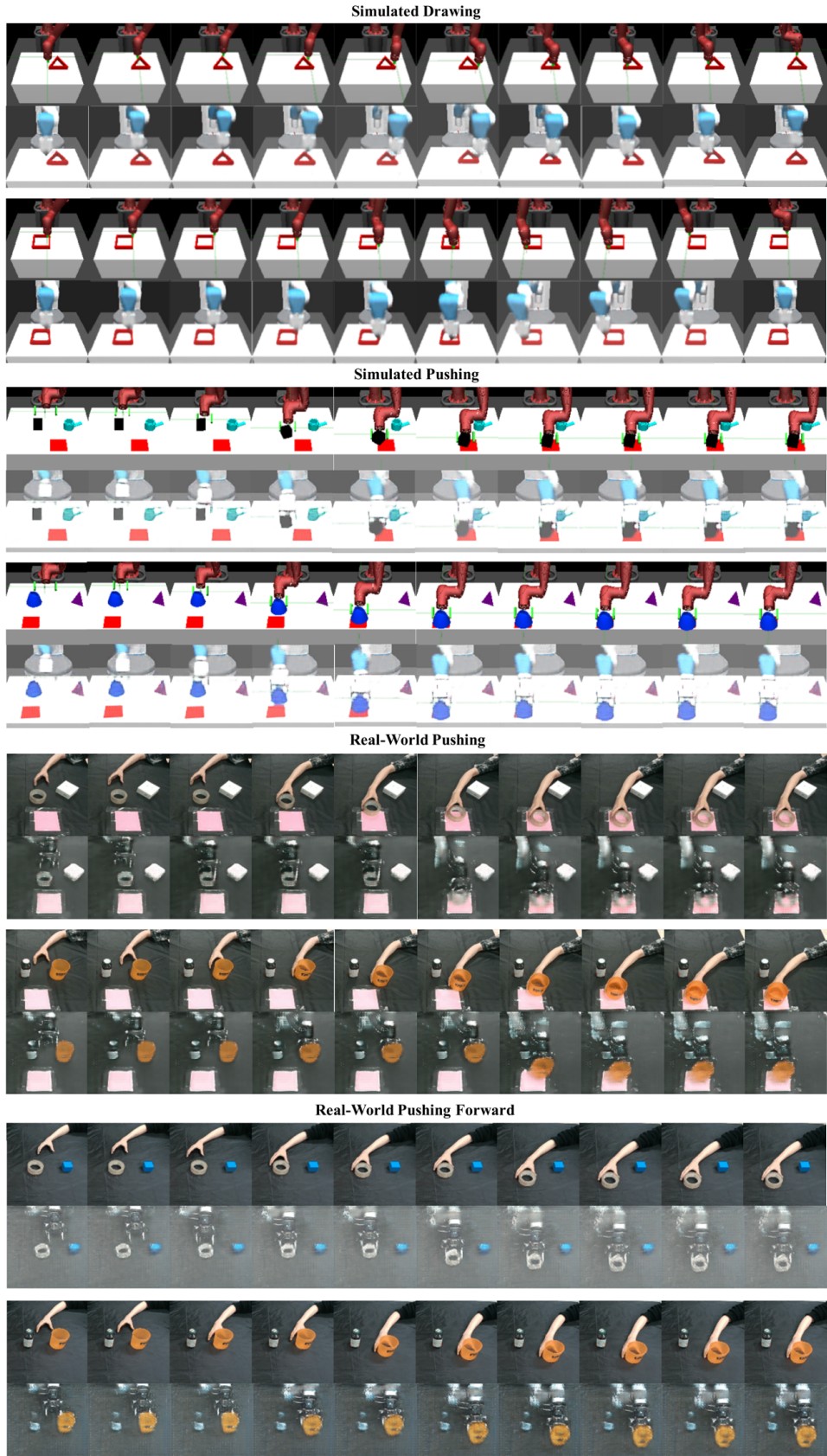

Figure 11: The examples of the translated videos on all tasks.

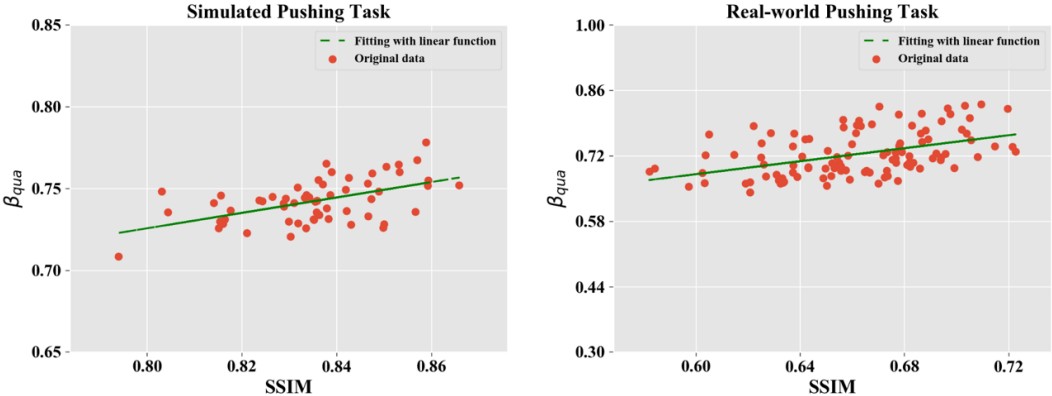

Figure 12: The correlation between the adaptive metric $\beta_{qua}$ and translation error $e$.

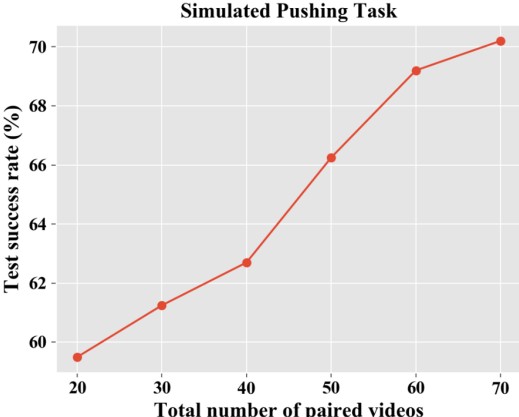

Figure 13: The test success rate on simulated pushing task as a function of the paired video numbers.

the number ups to more than 60, the growth of success rate substantially slows, which is why the amount of data utilized in our experiment is 60.

