# OpenReview forum: "Meta-Imitation Learning by Watching Video Demonstrations"
_ICLR.cc/2022/Conference — ICLR 2022 Poster_

### Official Review · Reviewer_EwC5 · 2021-11-01

**Correctness:** 4
**Technical Novelty And Significance:** 2
**Empirical Novelty And Significance:** 2
**Recommendation:** 6
**Confidence:** 4

**Main Review:**

Strengths:
- This method extends the previously proposed DAML algorithm, requiring only human video demonstrations (and also a small amount of paired data). This is appealing especially when it’s difficult to collect large amounts of demonstrations in the robot domain, e.g., through teleoperation. The resulting method performs comparably to the DAML algorithm, which requires more robot demonstration data.

- The experiments study the relevant ablations of the method, including using a decoupled CycleGAN and inverse dynamics model, using the translated images instead of the latent representation, and using a non-adaptive loss in DAML. The results show that all of the choices in their method make it stronger.

Weaknesses:
- The experiments lack comparisons to other methods besides DAML, which operates on a different set of assumptions. Since the success rates for all methods are quite low, it would be good to understand where the upper bound is. The method proposed in [1], like DAML, requires robot demonstrations as well, but may perform better than DAML. Another potential comparison is AVID [2]. Specifically, we can use all of the random robot data to train a visual dynamics model, and the rest of the data to train CycleGAN. One difference is that AVID solves multi-stage tasks and assumes the beginning of each stage is specified by the user; their reward function for planning is then defined by learned success classifiers. Instead, we could derive a simple dense reward function from the translated video, e.g., to match the translated video frame-by-frame or just to match the final frame as measured by the MSE.

- The SSIM and Brenner gradients are heuristic measures of the quality of the translated videos, and do not directly measure how accurate they are with respect to the true corresponding video in the robot domain. Since the meta-IL component operates on the latent representation and not the generated images themselves, it is an even more indirect measure. However, the non-adaptive loss does lead to worse results, so it empirically seems to work better. I’m curious how well these two heuristics actually correlate with the translation error. This could be evaluated on the small dataset of paired videos -- compute the MSE of the translated video and true robot video pair, and then measure the correlation with \beta_qua.

- It would also be nice to include some qualitative examples of the translated videos.

Other questions:
- The method still requires a small amount of paired human and robot data, which could consist of random actions (in their experiments, this is <= 100 trajectories). How does varying this number affect the performance of the method?

- The success rates for all methods are quite low. It’s also surprising that DAML, which leverages paired demonstrations, performs similarly to/worse than A-CycleGan + DAML, which doesn’t. How does the meta-policy learned by the different methods perform on the training shapes?

- Compared to DAML, the main mode of failure is due to control, i.e., not executing the right sequence of actions to complete the task, which may be explained by the prediction error in the inverse dynamics model. The chosen action space in the experiments is 3D - I’m curious how well it scales to higher dimensional action spaces, e.g., if the tasks required joint space control.


[1] Dasari et al. Transformers for One-Shot Visual Imitation. CoRL 2020.

[2] Smith et al. AVID: Learning Multi-Stage Tasks via Pixel-Level Translation of Human Videos. RSS 2020.


**Summary Of The Paper:**

This paper studies the meta-imitation learning from human videos problem, which aims to learn a new task by watching a few demonstrations of a human performing the task. While DAML [1], one of the relevant prior approaches to this problem, requires paired demonstrations in the human domain and robot domain for training, this approach only requires human videos and a small amount of paired data. The approach proposed in this paper translates human videos into the robot domain with a CycleGAN. Simultaneously, an inverse dynamics model is trained in the robot’s observation domain with randomly collected robot data to recover the actions from the translated demonstrations. Because the quality of the translations can be quite varied, higher-quality translations, measured based on the SSIM and Brenner gradient, are prioritized over lower-quality ones in the meta-imitation learning procedure, implemented by the DAML algorithm.

**Summary Of The Review:**

The paper tackles a challenging and compelling problem: meta-imitation learning of human demos with a small amount of paired robot demos. However, the experiments lack relevant comparisons to contextualize the significance of the proposed method. The heuristics used in the adaptive loss could also be better analyzed, e.g., by looking at their correlation with the MSE between the translated videos and true robot video.

---

> ### Author Response · Authors · 2021-11-20
> **Response to Reviewer #4  (Part 1/2)**
>
> We thank the reviewer for summarizing our work for tackling a “challenging and compelling” problem.  We have run additional experiments and enriched the paper. Below we hope to address the reviewers concerns:
>
> - Q1: The experiments lack comparisons to other methods besides DAML, ... Instead, we could derive a simple dense reward function from the translated video, e.g., to match the translated video frame-by-frame or just to match the final frame as measured by the MSE.
> > A1:  For the “the success rates for all methods are quite low”, we want to clarify this as the following two points:
> &emsp;1. The tasks designed in our experiments are challenging. As described in Experiment section, Para 2, for the shape-drawing skill, the robot needs to follow the predefined shape lines accurately, which may be easy when there is accurate action information but is hard to do when only has image information. For the pushing skill, the robot needs to push the target into the specific target area under the disturbance of never-seen distractors, which is more difficult than that designed in DAML paper where the robot only needs to push the target away under the disturbance of the same distractor as that in the demonstration. The experimental results are not very high. Please note that the results of DAML are also not very high, even with both human and robot demonstrations. We are sorry that the design of the challenging tasks misleads the reviewer thinks that the results are less powerful. To better show our powerful results, we supplement the experiment results of the real-world pushing task in the same condition as that in the DAML paper in Appendix B.3.  As shown in Table, we can see that DAML achieved a similar result (81.3\%) as that reported in DAML paper (88.9\%). While our method also achieved a comparable result (77.5\%) compared to DAML. We hope this would make our results more convincing.
> &emsp;2. The goal of this paper is to demonstrate that we can achieve com- parable results under the condition of only human demonstrations with the baseline which needs both human and robot demonstrations in equal task environments. Therefore, the key point is that we care more about whether we achieve comparable results with the baseline. It can be observed from experimental results that our method shows promising performance in visual imitation learning without laborious demonstration data collection.
> &emsp; For “comparisons to other methods”, the method in [1] is a good one-shot imitation learning work, even performs better than DAML, but it still requires robot demonstrations. Our method achieves similar performance with DAML, which means we find a solution to develop visual-imitation learning method only with human video demonstrations. Any method which needs robot demonstrations could be combined with our method to release their demand of demonstration data collection on imitation system while still guaranteeing the performance. As for AVID [2], whose fundamental method is model-based reinforcement learning, it requires interaction with the environment, whereas we do not. Then it would be improper for these two methods (AVID and Ours) to be tested under the same environment condition. In the next-step work, it would be a good choice for us to combine with reinforcement learning methods to improve the accuracy of tasks.
> | Method | Success Rate |
> | :-----| :----: |
> | DAML  | 81.3% |
> | **MILV (Ours)**  |77.5%  |
>
> - Q2: The SSIM and Brenner gradients are heuristic measures of the quality of the translated videos, and do not directly measure how accurate they are with respect to the true corresponding video in the robot domain. ... compute the MSE of the translated video and true robot video pair, and then measure the correlation with $\beta_{qua}$.
> > A2: Thanks for your inventive verification way. We suggest using the structural similarity index measure (SSIM) to compare paired data, which is more statistically sound. It considers comparison measurements of luminance, contrast and structure between the two images. We have calculated the SSIM of the translated video and true robot video pair as the true translation error $e$, and evaluate the correlation between the adaptive metric $\beta_{qua}$ and $e$ in Appendix B.5. The statistical result indicates that these two are positively correlated, which is in line with our original intention and intuition of the method.
>
> - Q3: It would also be nice to include some qualitative examples of the translated videos.
> > A3: Examples of translated videos corresponding to all tasks have been included in the Appendix B.4.
>
> ... continued in part 2/2.

---

> > ### Author Response · Authors · 2021-11-20
> > **Response to Reviewer #4 (Part 2/2)**
> >
> > - Q4: The method still requires a small amount of paired human and robot data, which could consist of random actions (in their experiments, this is $<=$100 trajectories). How does varying this number affect the performance of the method?
> > > A4: We have shown curves for the number of paired data points and test success rates on simulated pushing task in the Appendix B.6. The number of paired data we used for training in the paper is considered experimentally, achieving a trade-off between data collecting difficulties and experimental performance.
> >
> > - Q5: The success rates for all methods are quite low. It’s also surprising that DAML, which leverages paired demonstrations, performs similarly to/worse than A-CycleGan + DAML, which doesn’t. How does the meta-policy learned by the different methods perform on the training shapes?
> > > A5: Please refer to A1 for detailed clarification about the success rates and the comparison results.
> >
> >
> > - Q6: Compared to DAML, the main mode of failure is due to control, ...I’m curious how well it scales to higher dimensional action spaces, e.g., if the tasks required joint space control.
> > > A6: When scaling to higher dimensional action spaces, the overall structure of our algorithm will not be affected. While the policy spaces would be greatly increased, we need more research on finding a fast exploration method to search the optimal policy.
> >
> >
> > Thanks for your responsible and elaborate comments, which help us to support our work more fully. We are happy to continue iterating with your feedback.
> >
> > Reference:
> >
> > [1] Dasari et al. Transformers for One-Shot Visual Imitation. CoRL 2020.
> >
> > [2] Smith et al. AVID: Learning Multi-Stage Tasks via Pixel-Level Translation of Human Videos. RSS 2020.

---

> > > ### Comment · Reviewer_EwC5 · 2021-11-30
> > > **Thanks for your response**
> > >
> > > Thanks for the thorough response. It’s nice to see the analysis match the intuition behind the quality-based weighting.
> > >
> > > I agree that demonstrating similar performance to DAML is already a compelling result. Regarding the comparability to AVID, I believe the method proposed in your paper also requires interaction with the environment, i.e., the random robot videos used to trained the inverse dynamics model. So, I believe the data requirements are similar between your method and AVID (at least a variant that only trains the model on random interaction data).
> > >
> > > Most of my concerns were addressed, so I’ve updated my score.

---

### Official Review · Reviewer_337g · 2021-11-03

**Correctness:** 4
**Technical Novelty And Significance:** 3
**Empirical Novelty And Significance:** Not applicable
**Recommendation:** 8
**Confidence:** 3

**Main Review:**

Strengths

- This paper tackles a very challenging problem and accordingly presents an elaborately designed approach that brings together components from several state-of-the-art methods from slightly different areas. As a result, the overall approach is pretty novel.

Weaknesses

- The motivation of this paper is to reduce data collection complexity by only requiring human demonstrations, since robot demonstrations may require kinematic teaching / teleoperation which can be difficult to collect. However, while the proposed method doesn’t require robot demonstrations, it does require pairwise human and robot videos for training the CycleGAN component. My assumption is that this is also difficult to collect in practice, so I think the value of the proposed method is significantly diminished.

- In terms of the experiments, three tasks are presented, with two in simulation, where demonstrations are from a Fetch robot and the learner is a Sawyer robot, and one task in the real world that learns from human videos. While the simulated robot-to-robot experiments demonstrate some aspects of the algorithm, they are less powerful results. With only one task in the target setup, I feel the overall experimental results are not sufficiently rigorous.

- In addition, the final success rate achieved by the algorithm for the real-world pushing task is 56.3%, which is a lot lower than what DAML achieves as shown in the DAML paper (88.9%) on a highly similar task. While in the present paper, the proposed algorithm matches DAML’s performance (56.3 vs 58.8%), both success rates are low. I wonder if there are any structural limitations of the proposed approach that prevents it from achieving a higher rate. Consider running experiments with more training data; would the proposed algorithm still be able to match DAML’s performance?

- The writing of the paper needs significant polishing.

**Summary Of The Paper:**

This paper studies the problem of meta imitation learning of robot control policies from only video demonstrations of humans performing the tasks. The goal is to make data collection simpler by not requiring robot demonstrations which can be expensive to acquire. The method proposed by the authors combines multiple components. Specifically, a CycleGAN which translates human demo videos into robot demo videos, an Inverse Dynamics Model which infers robot action from consecutive video frames, a method to weigh the training loss according to the quality of translated images, and finally the same learning algorithm from DAML is used.

**Summary Of The Review:**

This paper studies a very challenging problem and presents a novel method, but the experiments are not convincing enough.

----

UPDATED the overall recommendation after authors' response. see details in thread.

---

> ### Author Response · Authors · 2021-11-20
> **Response to Reviewer #3**
>
> We thank the reviewer for characterizing our approach “pretty novel” and tackling a problem that is “very challenging”.  We hope to address any concerns with additional experiments updated in the paper and the following discussion.
>
> - Q1: The motivation of this paper is to reduce data collection complexity by only requiring human demonstrations, ....My assumption is that this is also difficult to collect in practice, so I think the value of the proposed method is significantly diminished.
> > A1: Although we require pairwise data, what we collect is robot random move data, that is, the robot does not need to interact with the environment in any elaborate way. With the cycle consistent restrictions of the CycleGAN model, our data does not have to be perfectly aligned, making collection much easier. On the other hand, these pairwise videos are just used to learn the content and style transfer between human domain and robot domain, so the number of paired videos required is small ($<=$110), far fewer than the number of robot demonstrations required in baseline ($>$600).
>
> - Q2:  In terms of the experiments, three tasks are presented, with two in simulation, ...With only one task in the target setup, I feel the overall experimental results are not sufficiently rigorous.
> > A2: In this paper, our goal is to explore the possibility for robots to learn just like humans, learning just by watching. The key point is learning by watching, whether from a human or another robot. So we design the experiments for robot-to-robot in simulation and human-to-robot in real-world to verify the generalization of the proposed method. For the “less powerful results” and “not sufficiently rigorous”, we want to clarify these as the following two points:
> &emsp;1. The tasks designed in our experiments are challenging. As described in Experiment section, Para 2, for the shape-drawing skill, the robot needs to follow the predefined shape lines accurately, which may be easy when there is accurate action information but is hard to do when only has image information. For the pushing skill, the robot needs to push the target into the specific target area under the disturbance of never-seen distractors, which is more difficult than that designed in DAML paper where the robot only needs to push the target away under the disturbance of the same distractor as that in the demonstration. The experimental results are not very high. Please note that the results of DAML are also not very high, even with both human and robot demonstrations. We are sorry that the design of the challenging tasks misleads the reviewer thinks that the results are less powerful. To better show our powerful results, we supplement the experiment results of the real-world pushing task in the same condition as that in the DAML paper in Appendix B.3.  As shown in Table, we can see that DAML achieved a similar result (81.3\%) as that reported in DAML paper (88.9\%). While our method also achieved a comparable result (77.5\%) compared to DAML. We hope this would make our results more convincing.
> &emsp;2. The goal of this paper is to demonstrate that we can achieve com- parable results under the condition of only human demonstrations with the baseline which needs both human and robot demonstrations in equal task environments. Therefore, the key point is that we care more about whether we achieve comparable results with the baseline. It can be observed from experimental results that our method shows promising performance in visual imitation learning without laborious demonstration data collection.
> | Method | Success Rate |
> | :-----| :----: |
> | DAML  | 81.3% |
> | **MILV (Ours)**  |77.5%  |
>
>
> - Q3: In addition, the final success rate achieved by the algorithm for the real-world pushing task is 56.3\%, ...Consider running experiments with more training data; would the proposed algorithm still be able to match DAML’s performance?
> > A3: As described in A2, to better show the performance of our method, we supplement the experiment results of a real-world pushing task designed in the same condition as that in DAML paper in Appendix B.3. Our method (77.5\%) and DAML (81.3\%) both achieved the similar result as that in DAML paper. Hope this results could address the reviewer’s concerns.
>
> - Q4: The writing of the paper needs significant polishing.
> > A4: We have updated the paper and in the revised version of our paper, we rearranged the logic and phrasing in the hope of better properly conveying our technique.  Sincerely we invite the reviewer to read the new one and give more valuable feedback.
>
> Thank you for your constructive review and please let us know if you have any more requested updates.

---

> > ### Comment · Reviewer_337g · 2021-11-28
> > **Response to authors' response**
> >
> > I thank the authors for the detailed response as above as well as additional experimental results.
> >
> > > we supplement the experiment results of the real-world pushing task in the same condition as that in the DAML paper in Appendix B.3. As shown in Table, we can see that DAML achieved a similar result (81.3%) as that reported in DAML paper (88.9%). While our method also achieved a comparable result (77.5%) compared to DAML
> >
> > These additional results alleviate the second and third weaknesses as in my original assessment.
> >
> > >these pairwise videos are just used to learn the content and style transfer between human domain and robot domain, so the number of paired videos required is small (110)
> >
> > More rigorously, an ablation study is needed to conclude the amount of such pairwise data required and if that in any ways correlates with the task complexity. But as experimentally demonstrated, the amount required indeed doesn't seem high.
> >
> > -----
> >
> > Given the second and third weaknesses in my original response are my top concerns, now that they were addressed, I have updated my overall recommendation score from 5 to 8.

---

### Official Review · Reviewer_tm35 · 2021-11-05

**Correctness:** 3
**Technical Novelty And Significance:** 3
**Empirical Novelty And Significance:** 3
**Recommendation:** 8
**Confidence:** 3

**Main Review:**

+ The task is very relevant in the robotics community
+ The paper is well structured
+ The experimental analysis is convincing

- The reviewer finds some part a bit unclear. Here is a tentative summary of the main concerns:
   * Not fully clear to the reviewer how supervision works in A-CycleGAN (on the robot side)
   * Not fully clear to the reviewer how information on action and on environment is disentangled and exploited (Sec. 4.1)
   * The reviewer fails in gathering an intuition about the interplay between the different modules in the architecture. In particular, the Inverse
      Dynamic Model lacks details, and the Meta-Imitation Learning, one of the core parts of the work, has a description which is a but
      compressed.

**Summary Of The Paper:**

The paper considers the problem of robot imitation learning from video demonstration, without the need of accessing to an explicit representation of the action of the demonstrator.
The authors introduce an approach for meta-imitation learning that aims to traslate human videos to robot demonstrations from a single video demonstration. The correspondence between human and robot is learned end-to-end exploiting a generative architectures, A-CycleGAN, extension of CycleGAN. The method is experimentally assessed on different visual tasks.

**Summary Of The Review:**

I find the topic very interesting and the video-based approach is certainly among the most relevant in robotics. However, I find the clarity of the presentation could be improved to fully appreciate the essence of the approach

---

> ### Author Response · Authors · 2021-11-20
> **Response to Reviewer #2**
>
> We thank the reviewer for convincing the structure and experimental analysis of our paper.  We have undergone a thorough revision of the paper to deliver a clearer presentation of the method.
>
> - Q1: Not fully clear to the reviewer how supervision works in A-CycleGAN (on the robot side)
> > A1: The supervision work of the A-CycleGAN on the robot domain is the process of training the inverse dynamic model using the actual random motion data of the robot.  Specifically, the inverse dynamic model takes as input current visual observation of the robot $x_t^r$ along with the observation for the next time step $x_{t+1}^r$ to predict the actions $a_t^r = {\pi _I}(x_t^r,x_\{t + 1\}^r)$. We use robot random move video $(x_1^r, a_1^r, ...,x_T^r,a_T^r)$ as supervised data because it can be generated by self-supervision and contains available observation and action of the robot domain.
> - Q2:  Not fully clear to the reviewer how information on action and on environment is disentangled and exploited (Sec. 4.1)
> > A2: The A-CycleGAN can be disentangled into two parts, the CycleGAN part and the Inverse Dynamic Model part. When employed, the human demonstrations are translated to the robot domain by translator $G$, and then the corresponding actions are predicted through the inverse dynamic model.These two parts share the network structure of encoding layers of generator $F$, but they are independent while inferring.
> - Q3: The reviewer fails in gathering an intuition about the interplay between the different modules in the architecture. In particular, the Inverse Dynamic Model lacks details, and the Meta-Imitation Learning, one of the core parts of the work, has a description which is a but compressed.
> > A3: In our paper, the proposed architecture comprises two main modules: A-CycleGAN and meta-imitation learning module, in which the A-CycleGAN module provides the stable input and supervision information for the following meta-imitation learning module.
> &emsp;
> > As described in A1, the inverse dynamic model is built to predict the robot actions based on the observations and is trained parallelly with the CycleGAN. The detailed structure of the inverse controller can be found in Appendix A.2. We introduce the inverse dynamic model to obtain the necessary robot action information when only access to visual observation. We have revised the paper in Section 4.1 to reflect a detailed description.
> &emsp;
> > Concerning the Meta-Imitation Learning section, the fundamental theory of our approach is based on DAML [1] which is introduced in Preliminaries, Section 3.2. And we also recommend the paper [1] for more details.
>
> Thanks for your interest in our work and help us to revise the detailed description in the paper. Please let us know if you have any additional comments or questions.
>
> Reference:
>
> [1] Tianhe Yu, Chelsea Finn, Annie Xie, Sudeep Dasari, Tianhao Zhang, Pieter Abbeel, and Sergey Levine. One-shot imitation from observing humans via domain-adaptive meta-learning. In Robotics: Science and Systems 2018, 2018.

---

> > ### Comment · Reviewer_tm35 · 2021-11-29
> > **Response to authors**
> >
> > I thank the authors for the feedback and I appreciate their effort in clarifying the aspects on which I raised issues. I'll change the score accordingly

---

### Official Review · Reviewer_9GZg · 2021-11-07

**Correctness:** 4
**Technical Novelty And Significance:** 3
**Empirical Novelty And Significance:** 3
**Recommendation:** 8
**Confidence:** 4

**Main Review:**

Strengths:
- Paper is well-written.
- The paper only uses human video demonstration and no robot demo/states and actions to learn a task seems unique to me.
- Paper situates the problem very well and presents good series of prior work to solve the problem addressed in the paper.
- Good range of experiments

Weakness:
- In the introduction section, the authors mention about challenging tasks, could you please explain in one line what is the difficulty of the tasks that you are attempting the robot to learn?
- It will be good add a description of "imagined videos" that has been reference in the paper multiple times. Maybe add in the Method section.
- I encourage the authors to submit a video of experiment results for both simulated tasks and real-world pushing task.
- A question asked in the Introduction section para 2, is not clear to me. Could you please re-write the sentence with more clarity?

**Summary Of The Paper:**

The paper talks about solving an interesting problem of meta-imitaion learning from watching human video demonstration. The paper proposes a A-CycleGAN approach to learn a latent representation for human-robot video correspondence and a self-adaptive meta imitation learning under imagined latent space to evaluate the quality of translated data. The proposed methods in the paper achieves comparable performance to the state-of-the-art method.

**Summary Of The Review:**

Based on paper methodology contributions and above merits (explained in Main Review section).

---

> ### Author Response · Authors · 2021-11-20
> **Response to Reviewer #1**
>
> We thank the reviewer for characterizing our paper “well written” and tackling a "unique" problem. We hope to provide explanation and the iteration of the paper to make the work clearer.
>
> - Q1: Please explain in one line what is the difficulty of the tasks that you are attempting the robot to learn?
> > A1: The main difficulty is that on the condition of only human video demonstrations without access to the actions of the demonstrator, tasks need the robot to recover the right trajectories under diverse scenes. For example, in the shape-drawing task, the robot needs to follow the given shape lines accurately, which may be easy when there is accurate action information but is hard to do with only image information. In the pushing task, there are distinct target objects, distractors, and positions, which is challenging for the robot to recognize the target and execute the actions under the disturbance of never-seen distractors.
>
> - Q2: It will be good add a description of "imagined videos" that has been reference in the paper multiple times. Maybe add in the Method section.
> > A2: We have added its description to the para 1 of Section 4.1. Indeed, the "imagined videos" refer to the robot videos translated from the human demonstrations by the A-CycleGAN.
>
> - Q3: I encourage the authors to submit a video of experiment results for both simulated tasks and real-world pushing task.
> > A3: Thanks for the suggestion. We will submit the video of experiment results soon.
>
> - Q4: A question asked in the Introduction section para 2, is not clear to me. Could you please re-write the sentence with more clarity?
> > A4: We are really sorry that our previous description may not be accurate enough.  We have clarified this statement. The following sentences have been reorganized:
>       - Motivated by this, we aim to endow robots with the ability to learn manipulation skills via video demonstrations from humans without access to the actions of the demonstrator.  Then the key challenge we would face is how to bridge the human-robot domain gap caused by the morphological difference and infer the performed skills from the raw video.
>
> Thanks for your detailed review. Please let us know if there are any outstanding concerns or desired clarifications as we continue to update the paper.

---

> > ### Author Response · Authors · 2021-11-22
> > **Supplementary video**
> >
> > Thanks for your reviewer. We have uploaded the video to the supplementary material.

---

### Decision · Program_Chairs · 2022-01-20

**Decision:**

Accept (Poster)

**Comment:**

This is an interesting paper working on the difficult problem of learning from video demonstration. The authors provided convincing experimental solutions for visual representation, domain adaptation, and imitation. It would be a nice ICLR paper.